# Can the Brain Do Backpropagation?
# — Exact Implementation of Backpropagation in Predictive Coding Networks

**Yuhang Song**[1], **Thomas Lukasiewicz**[1], **Zhenghua Xu**[2,*], **Rafal Bogacz**[3]
[1]Department of Computer Science, University of Oxford, UK
[2]State Key Laboratory of Reliability and Intelligence of Electrical Equipment,
Hebei University of Technology, Tianjin, China
[3]MRC Brain Network Dynamics Unit, University of Oxford, UK
`yuhang.song@some.ox.ac.uk, thomas.lukasiewicz@cs.ox.ac.uk,`
`zhenghua.xu@hebut.edu.cn, rafal.bogacz@ndcn.ox.ac.uk`

## Abstract

Backpropagation (BP) has been the most successful algorithm used to train artificial neural networks. However, there are several gaps between BP and learning in biologically plausible neuronal networks of the brain (learning in the brain, or simply BL, for short), in particular, (1) it has been unclear to date, if BP can be implemented exactly via BL, (2) there is a lack of local plasticity in BP, i.e., weight updates require information that is not locally available, while BL utilizes only locally available information, and (3) there is a lack of autonomy in BP, i.e., some external control over the neural network is required (e.g., switching between prediction and learning stages requires changes to dynamics and synaptic plasticity rules), while BL works fully autonomously. Bridging such gaps, i.e., understanding how BP can be approximated by BL, has been of major interest in both neuroscience and machine learning. Despite tremendous efforts, however, no previous model has bridged the gaps at a degree of demonstrating an equivalence to BP, instead, only approximations to BP have been shown. Here, we present for the first time a framework within BL that bridges the above crucial gaps. We propose a BL model that (1) produces *exactly the same* updates of the neural weights as BP, while (2) employing local plasticity, i.e., all neurons perform only local computations, done simultaneously. We then modify it to an alternative BL model that (3) also works fully autonomously. Overall, our work provides important evidence for the debate on the long-disputed question whether the brain can perform BP.

## 1   Introduction

Backpropagation (BP) [1–3] as the main principle underlying learning in deep artificial neural networks (ANNs) [4] has long been criticized for its biological implausibility (i.e., BP's computational procedures and principles are unrealistic to be implemented in the brain) [5–10]. Despite such criticisms, growing evidence demonstrates that ANNs trained with BP outperform alternative frameworks [11], as well as closely reproduce activity patterns observed in the cortex [12–20]. As indicated in [10, 21], since we apparently cannot find a better alternative than BP, the brain is likely to employ at least the core principles underlying BP, but perhaps implements them in a different way. Hence, bridging the gaps between BP and learning in biological neuronal networks of the brain (learning in the brain, for short, or simply BL) has been a major open question for both neuroscience and machine

---

learning [10, 21–29]. Such gaps are reviewed in [30–33], the most crucial and most intensively studied gaps are (1) that it has been unclear to date, if BP can be implemented exactly via BL, (2) BP's lack of local plasticity, i.e., weight updates in BP require information that is not locally available, while in BL weights are typically modified only on the basis of activities of two neurons (connected via synapses), and (3) BP's lack of autonomy, i.e., BP requires some external control over the network (e.g., to switch between prediction and learning), while BL works fully autonomously.

Tremendous research efforts aimed at filling these gaps, trying to approximate BP in BL models. However, earlier BL models were not scaling to larger and more complicated problems [8, 34–42]. More recent works show the capacity of scaling up BL to the level of BP [43–57]. However, to date, none of the earlier or recent models has bridged the gaps at a degree of demonstrating an equivalence to BP, though some of them [37, 48, 53, 58–60] demonstrate that they approximate BP, or are equivalent to BP under unrealistic restrictions, e.g., the feedback is sufficiently weak [61, 48, 62]. The unability to fully close the gaps between BP and BL is keeping the community's concerns open, questioning the link between the power of artificial intelligence and that of biological intelligence.

Recently, an approach based on predictive coding networks (PCNs), a widely used framework for describing information processing in the brain [31], has partially bridged these crucial gaps. This model employed a supervised learning algorithm for PCNs to which we refer as inference learning (IL) [48]. IL is capable of approximating BP with attractive properties: local plasticity and autonomous switching between prediction and learning. However, IL has not yet fully bridged the gaps: (1) IL is only an approximation of BP, rather than an equivalent algorithm, and (2) it is not fully autonomous, as it still requires a signal controlling when to update weights.

Therefore, in this paper, we propose the first BL model that is equivalent to BP while satisfying local plasticity and full autonomy. The main contributions of this paper are briefly summarized as follows:

- To develop a BL approach that is equivalent to BP, we propose three easy-to-satisfy conditions for IL, under which IL produces exactly the same weight updates as BP. We call the proposed approach *zero-divergent IL (Z-IL)*. In addition to being equivalent to BP, Z-IL also satisfies the properties of local plasticity and partial autonomy between prediction and learning.

- However, Z-IL is still not fully autonomous, as weight updates in Z-IL still require a triggering signal. We thus further propose a *fully autonomous Z-IL (Fa-Z-IL)* model, which requires *no* control signal anymore. Fa-Z-IL is the first BL model that not only produces exactly the same weight updates as BP, but also performs all computations *locally*, *simultaneously*, and *fully autonomously*.

- We prove the general result that Z-IL is equivalent to BP while satisfying local plasticity, and that Fa-Z-IL is additionally fully autonomous. Consequently, this work may bridge the crucial gaps between BP and BL, thus, could provide previously missing evidence to the debate on whether BP could describe learning in the brain, and links the power of biological and machine intelligence.

The rest of this paper is organized as follows. Section 2 recalls BP in ANNs and IL in PCNs. In Section 3, we develop Z-IL in PCNs and show its equivalence to BP and its local plasticity. Section 4 focuses on how Z-IL can be realized with full autonomy. Sections 5 and 7 provide some further experimental results and a conclusion, respectively.

## 2  Preliminaries

We now briefly recall artificial neural networks (ANNs) and predictive coding networks (PCNs), which are trained with backpropagation (BP) and inference learning (IL), respectively. For both models, we describe two stages, namely, (1) the prediction stage, when no supervision signal is present, and the goal is to use the current parameters to make a prediction, and (2) the learning stage, when a supervision signal is present, and the goal is to update the current parameters. Following [48], we use a slightly different notation than in the original descriptions to highlight the correspondence between the variables in the two models. The notation is summarized in Table 1 and will be introduced in detail as the models are described. To make the dimension of variables explicit, we denote vectors with a bar (e.g., $\overline{x} = (x_1, x_2, \ldots, x_n)$ and $\overline{0} = (0, 0, \ldots, 0)$).

Table 1: Notation for ANNs and PCNs.

| | Value-node activity | Predicted value-node activity | Error term or node | Weight | Objective function | Activation function | Layer size | Number of layers | Input signal | Supervision signal | Learning rate for weights | Integration step for inference |
|---|---|---|---|---|---|---|---|---|---|---|---|---|
| ANNs | $y_i^l$ | – | $\delta_i^l$ | $w_{i,j}^l$ | $E$ | $f$ | $n^l$ | $l_{\max}+1$ | $s_i^{\text{in}}$ | $s_i^{\text{out}}$ | $\alpha$ | – |
| PCNs | $x_{i,t}^l$ | $\mu_{i,t}^l$ | $\varepsilon_{i,t}^l$ | $\theta_{i,j}^l$ | $F_t$ | | | | | | | $\gamma$ |

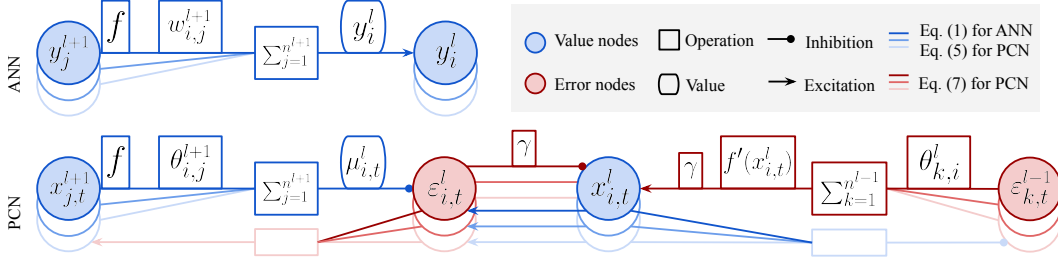

Figure 1: ANNs and PCNs trained with BP and IL, respectively.

## 2.1 ANNs trained with BP

*Artificial neural networks* (*ANNs*) [3] are organized in layers, with multiple neuron-like value nodes in each layer. Following [48], to make the link to PCNs more visible, we change the direction in which layers are numbered and index the output layer by $0$ and the input layer by $l_{\max}$. We denote by $y_i^l$ the input to the $i$-th node in the $l$-th layer. Thus, the connections between adjacent layers are:

$$y_i^l = \sum_{j=1}^{n^{l+1}} w_{i,j}^{l+1} f(y_j^{l+1}), \tag{1}$$

where $f$ is the activation function, $w_{i,j}^{l+1}$ is the weight from the $j^{th}$ node in the $(l+1)^{th}$ layer to the $i^{th}$ node in the $l^{th}$ layer, and $n^{l+1}$ is the number of nodes in layer $(l+1)$. Note that, in this paper, we consider only the case where there are only weights as parameters and no bias values. However, all results of this paper can be easily extended to the case with bias values as additional parameters; see supplementary material.

**Prediction:** Given values of the input $\bar{s}^{\text{in}} = (s_1^{\text{in}}, \dots, s_{n^{l_{\max}}}^{\text{in}})$, every $y_i^{l_{\max}}$ in the ANN is set to the corresponding $s_i^{\text{in}}$, and then every $y_i^0$ is computed as the prediction via Eq. (1).

**Learning:** Given a pair $(\bar{s}^{\text{in}}, \bar{s}^{\text{out}})$ from the training set, $\bar{y}^0 = (y_1^0, \dots, y_{n^0}^0)$ is computed via Eq. (1) from $\bar{s}^{\text{in}}$ as input and compared with $\bar{s}^{\text{out}}$ via the following objective function $E$:

$$E = \tfrac{1}{2} \sum_{i=1}^{n^0} (s_i^{\text{out}} - y_i^0)^2. \tag{2}$$

*Backpropagation* (*BP*) updates the weights of the ANN by:

$$\Delta w_{i,j}^{l+1} = -\alpha \cdot \partial E / \partial w_{i,j}^{l+1} = \alpha \cdot \delta_i^l f(y_j^{l+1}), \tag{3}$$

where $\alpha$ is the learning rate, and $\delta_i^l = \partial E / \partial y_i^l$ is the *error term*, given as follows:

$$\delta_i^l = \begin{cases} s_i^{\text{out}} - y_i^0 & \text{if } l = 0 \,; \\ f'(y_i^l) \sum_{k=1}^{n^{l-1}} \delta_k^{l-1} w_{k,i}^l & \text{if } l \in \{1, \dots, l_{\max} - 1\} \,. \end{cases} \tag{4}$$

## 2.2 PCNs trained with IL

*Predictive coding networks* (*PCNs*) [31] are a widely used model of information processing in the brain, originally developed for unsupervised learning. It has recently been shown that when a PCN

is used for supervised learning, it closely approximates BP [48]. As the learning algorithm in [48] involves inferring the values of hidden nodes, we call it *inference learning* (*IL*). Thus, we denote by $t$ the time axis during inference. As shown in Fig. 1, a PCN contains value nodes (blue nodes, with the activity of $x_{i,t}^l$), which are each associated with corresponding prediction-error nodes (error nodes: red nodes, with the activity of $\varepsilon_{i,t}^l$). Differently from ANNs, which propagate the activity between value nodes directly, PCNs propagate the activity between value nodes $x_{i,t}^l$ via the error nodes $\varepsilon_{i,t}^l$:

$$\mu_{i,t}^l = \sum_{j=1}^{n^{l+1}} \theta_{i,j}^{l+1} f(x_{j,t}^{l+1}) \quad \text{and} \quad \varepsilon_{i,t}^l = x_{i,t}^l - \mu_{i,t}^l, \tag{5}$$

where the $\theta_{i,j}^{l+1}$'s are the connection weights, paralleling $w_{i,j}^{l+1}$ in the described ANN, and $\mu_{i,t}^l$ denotes the prediction of $x_{i,t}^l$ based on the value nodes in a higher layer $x_{j,t}^{l+1}$. Thus, the error node $\varepsilon_{i,t}^l$ computes the difference between the actual and the predicted $x_{i,t}^l$. The value node $x_{i,t}^l$ is modified so that the overall energy $F_t$ in $\varepsilon_{i,t}^l$ is minimized all the time:

$$F_t = \sum_{l=0}^{l_{\max}-1} \sum_{i=1}^{n^l} \tfrac{1}{2} (\varepsilon_{i,t}^l)^2 . \tag{6}$$

In this way, $x_{i,t}^l$ tends to move close to $\mu_{i,t}^l$. Such a process of minimizing $F_t$ by modifying all $x_{i,t}^l$ is called *inference*, and it is running during both prediction and learning. Inference minimizes $F_t$ by modifying $x_{i,t}^l$, following a unified rule for both stages:

$$\Delta x_{i,t}^l = \begin{cases} 0 & \text{if } l = l_{\max} \\ \gamma \cdot (-\varepsilon_{i,t}^l + f'(x_{i,t}^l) \sum_{k=1}^{n^{l-1}} \varepsilon_{k,t}^{l-1} \theta_{k,i}^l) & \text{if } l \in \{1, \dots, l_{\max}-1\} \\ \gamma \cdot (-\varepsilon_{i,t}^l) & \text{if } l = 0 \text{ during prediction} \\ 0 & \text{if } l = 0 \text{ during learning,} \end{cases} \tag{7}$$

where $x_{i,t+1}^l = x_{i,t}^l + \Delta x_{i,t}^l$, and $\gamma$ is the integration step for $x_{i,t}^l$. Here, $\Delta x_{i,t}^l$ is different between prediction and learning only for $l = 0$, as the output value nodes $x_{i,t}^0$ are left unconstrained during prediction and are fixed to $s_i^{\text{out}}$ during learning. $\Delta x_{i,t}^l$ is zero for $l = l_{\max}$, as $x_{i,t}^{l_{\max}}$ is fixed to $s_i^{\text{in}}$ in both stages. Eqs. (5) and (7) can be evaluated in a network of simple neurons, as illustrated in Fig. 1.

**Prediction:** Given an input $\bar{s}^{\text{in}}$, the value nodes $x_{i,t}^{l_{\max}}$ in the input layer are set to $s_i^{\text{in}}$. Then, all the error nodes $\varepsilon_{i,t}^l$ are optimized by the inference process and decay to zero as $t \to \infty$. Thus, the value nodes $x_{i,t}^l$ converge to $\mu_{i,t}^l$, the same values as $y_i^l$ of the corresponding ANN with the same weights.

**Learning:** Given a pair $(\bar{s}^{\text{in}}, \bar{s}^{\text{out}})$ from the training set, the value nodes of both the input and the output layers are set to the training pair (i.e., $x_{i,t}^{l_{\max}} = s_i^{\text{in}}$ and $x_{i,t}^0 = s_i^{\text{out}}$); thus,

$$\varepsilon_{i,t}^0 = x_{i,t}^0 - \mu_{i,t}^0 = s_i^{\text{out}} - \mu_{i,t}^0. \tag{8}$$

Optimized by the inference process, the error nodes $\varepsilon_{i,t}^l$ can no longer decay to zero; instead, they converge to values as if the errors had been backpropagated. Once the inference converges to an equilibrium ($t = t_c$), where $t_c$ is a fixed large number, a *weight update* is performed. The weights $\theta_{i,j}^{l+1}$ are updated to minimize the same objective function $F_t$; thus,

$$\Delta \theta_{i,j}^{l+1} = -\alpha \cdot \partial F_t / \partial \theta_{i,j}^{l+1} = \alpha \cdot \varepsilon_{i,t}^l f(x_{j,t}^{l+1}), \tag{9}$$

where $\alpha$ is the learning rate. By Eqs. (7) and (9), all computations are local (local plasticity) in IL, and, as stated, the model can autonomously switch between prediction and learning (some autonomy), via running inference. However, a control signal is still needed to trigger the weights update at $t = t_c$; thus, full autonomy is not realized yet. The learning of IL is summarized in Algorithm 1. Note that detailed derivations of Eqs. (7) and (9) are given in the supplementary material (and in [48]).

## 3   IL with Zero Divergence from BP and with Local Plasticity

We now first describe the temporal training dynamics of BP in ANNs and IL in PCNs. Based on this, we then propose IL in PCNs with zero divergence from BP (and local plasticity), called Z-IL.

**Temporal Training Dynamics.**   We first describe the temporal training dynamics of BP in ANNs and IL in PCNs; see Fig. 2. We assume that we train the networks on a pair $(\bar{s}^{\text{in}}, \bar{s}^{\text{out}})$ from the dataset, which is presented for a period of time $T$, before it is changed to another pair, moving to the next

---
**Algorithm 1** Learning one training pair $(\overline{s}^{\text{in}}, \overline{s}^{\text{out}})$ (presented for the duration $T$) with IL
---
**Require:** $\overline{x}_0^{l_{\max}}$ is fixed to $\overline{s}^{\text{in}}$, $\overline{x}_0^0$ is fixed to $\overline{s}^{\text{out}}$.
 1: **for** $t = 0$ to $T$ **do**                                              // presenting a training pair
 2:     **for** each neuron $i$ in each level $l$ **do**                   // in parallel in the brain
 3:         Update $x_{i,t}^l$ to minimize $F_t$ via Eq. (7)             // inference
 4:         **if** $t = t_c$ **then**                                // external control signal
 5:             Update each $\theta_{i,j}^{l+1}$ to minimize $F_t$ via Eq. (9)
 6:             **return**                                // the brain rests
 7:         **end if**
 8:     **end for**
 9: **end for**
---

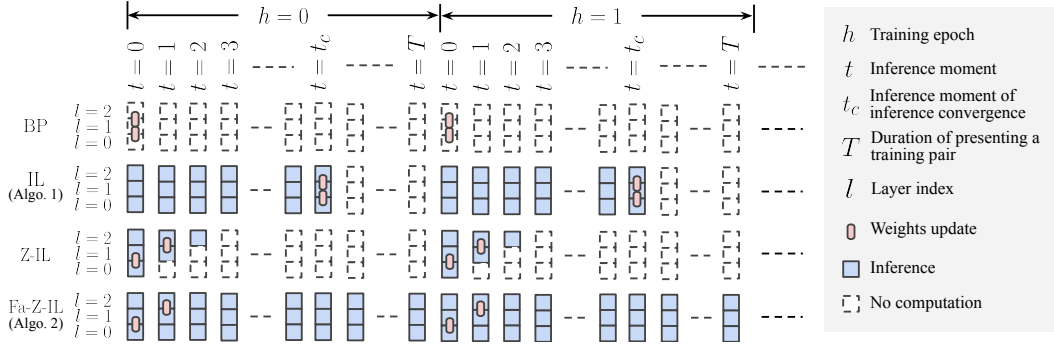

Figure 2: Comparison of the temporal training dynamics of BP, IL, Z-IL, and Fa-Z-IL. We assume that we train the networks on a pair $(\overline{s}^{\text{in}}, \overline{s}^{\text{out}})$ from the dataset, which is presented for a period of time $T$, before it is changed to another pair, moving to the next training epoch $h$. Within a single training epoch $h$, $(\overline{s}^{\text{in}}, \overline{s}^{\text{out}})$ stays unchanged, and $t$ runs from 0. As stated before, $t$ is the time axis during inference, which means that IL (also Z-IL and Fa-Z-IL) in PCNs run inference starting from $t = 0$. The squares and rounded rectangles represent nodes in one layer and connection weights between nodes in two layers of a neural network, respectively: BP (first row) only conducts weights updates in one training epoch, while IL (second row) conducts inference until it converges ($t = t_c$) and updates weights (assuming $T \geq t_c$). Note that C1 is omitted in the figure for simplicity.

training epoch $h$. Within a single training epoch $h$, $(\overline{s}^{\text{in}}, \overline{s}^{\text{out}})$ stays unchanged, and $t$ runs from 0. As stated before, $t$ is the time axis during inference, which means that IL (also Z-IL and Fa-Z-IL, proposed below) in PCNs run inference starting from $t = 0$. In Fig. 2, squares and rounded rectangles represent nodes in one layer and connection weights between nodes in two layers of a neural network, respectively: BP (first row) only conducts weights updates in one training epoch, while IL (second row) conducts inference until it converges ($t = t_c$) and updates weights (assuming $T \geq t_c$).

In the rest of this section and in Section 4, we introduce *zero-divergent IL (Z-IL)* and *fully autonomous zero-divergent IL (Fa-Z-IL)* in PCNs, respectively: Z-IL (third row) also conducts inference but until specific inference moments $t = l$ and updates weights between the layers $l$ and $l + 1$, while Fa-Z-IL (fourth row) conducts inference all the time and weights update is trigged autonomously at the same inference moments as Z-IL.

**Zero-divergent IL.** We now present three conditions C1 to C3 under which IL in a PCN, denoted *zero-divergent IL (Z-IL)*, produces exactly the same weights as BP in the corresponding ANN (having the same initial weights as the given PCN) applied to the same datapoints $s = (\overline{s}^{\text{in}}, \overline{s}^{\text{out}})$. In the following, we first describe the three conditions C1 to C3, and then formally state (and prove in the supplementary material) that Z-IL produces exactly the same weights as BP.

**C1:** Every $x_{i,t}^l$ and every $\mu_{i,t}^l$, $l \in \{1, \ldots, l_{\max}-1\}$, at $t=0$ is equal to $y_i^l$ in the corresponding ANN with input $\overline{s}^{\text{in}}$. In particular, this also implies that $\varepsilon_{i,t}^l = 0$ at $t=0$, for $l \in \{1, \ldots, l_{\max}-1\}$. This condition is naturally satisfied in PCNs, if before the start of each training epoch over a training pair

---

**Algorithm 2** Learning one training pair $(\overline{s}^{\text{in}}, \overline{s}^{\text{out}})$ (presented for the duration $T$) with Fa-Z-IL

---

**Require:** $\overline{x}_0^{l_{\max}}$ is fixed to $\overline{s}^{\text{in}}$, $\overline{x}_0^0$ is fixed to $\overline{s}^{\text{out}}$.
**Require:** $x_{i,0}^l = \mu_{i,0}^l$ for $l \in \{1, \ldots, l_{\max} - 1\}$ (C1), and $\gamma = 1$ (C3).
 1: **for** $t = 0$ to $T$ **do**                                    // presenting a training pair
 2:  **for** each neuron $i$ in each level $l$ **do**               // in parallel in the brain
 3:    Update $x_{i,t}^l$ to minimize $F_t$ via Eq. (7)          // inference
 4:    Update each $\theta_{i,j}^{l+1}$ to minimize $F_t$ via Eq. (9) with learning rate $\alpha \cdot \phi(\varepsilon_{i,t}^l)$
 5:  **end for**
 6: **end for**

---

$(\overline{s}^{\text{in}}, \overline{s}^{\text{out}})$, the input $\overline{s}^{\text{in}}$ has been presented, and the network has converged in the prediction stage (see Section 2.2). This condition corresponds to a requirement in BP, that it needs one forward pass from $\overline{s}^{\text{in}}$ to compute the prediction before conducting weights updates with the supervision signal $\overline{s}^{\text{out}}$. Note that neither the forward pass for BP nor this initialization for IL are shown in Fig. 2. Note that this condition is also applied in [48].

**C2:** Every weight $\theta_{i,j}^{l+1}$, $l \in \{0, \ldots, l_{\max} - 1\}$, is updated at $t = l$, that is, at a very specific inference moment, related to the layer that the weight belongs to. This may seem quite strict, but it can actually be implemented with full autonomy (see Section 4).

**C3:** The integration step of inference $\gamma$ is set to $1$. Note that solely relaxing this condition (keeping C1 and C2 satisfied) results in BP with a different learning rate for different layers, where $\gamma$ is the decay factor of this learning rate along layers (see supplementary material).

To prove the above equivalence statement under C1 to C3, we develop two theorems in order (proved in the supplementary material). The following first theorem formally states that the prediction error in the PCN with IL on $s$ under C1 to C3 is equal to the error term in its ANN with BP on $s$.

**Theorem 3.1.** *Let $M$ be a PCN, $M'$ be its corresponding ANN (with the same initial weights as $M$), and let $s$ be a datapoint. Then, every prediction error $\varepsilon_{i,t}^l$ at $t = l$, $l \in \{0, \ldots, l_{max} - 1\}$, in $M$ trained with IL on $s$ under C1 and C3 is equal to the error term $\delta_i^l$ in $M'$ trained with BP on $s$.*

We next formally state that every weights update in the PCN with IL on $s$ under C1 to C3 is equal to the weights update in its ANN with BP on $s$. This then immediately implies that the final weights of the PCN with IL on $s$ under C1 to C3 are equal to the final weights of its ANN with BP on $s$.

**Theorem 3.2.** *Let $M$ be a PCN, $M'$ be its corresponding ANN (with the same initial weights as $M$), and let $s$ be a datapoint. Then, every update $\Delta\theta_{i,j}^{l+1}$ at $t = l$, $l \in \{0, \ldots, l_{max} - 1\}$, in $M$ trained with IL on $s$ under C1 and C3 is equal to the update $\Delta w_{i,j}^{l+1}$ in $M'$ trained with BP on $s$.*

## 4 Z-IL with Full Autonomy

Both IL and Z-IL in PCNs optimize the unified objective function of Eq. (6) during prediction and learning. Thus, they both enjoy some autonomy already. Specifically, they can autonomously switch between prediction and learning by running inference, depending only on whether $\overline{x}_t^0$ is fixed to $\overline{s}^{\text{out}}$ or left unconstrained. However, when to update the weights still requires the control signal at $t = t_c$ and $t = l$ for IL and Z-IL, respectively.

Targeting at removing this control signal from Z-IL, we now propose *Fa-Z-IL*, which realizes Z-IL with full autonomy. Specifically, we propose a function $\phi(\cdot)$, which takes in $\varepsilon_{i,t}^l$ and modulates the learning rate $\alpha$, producing a local learning rate for $\theta_{i,j}^{l+1}$. As can be seen from Fig. 1, $\varepsilon_{i,t}^l$ is directly connected to $\theta_{i,j}^{l+1}$, meaning that $\phi(\cdot)$ works locally at a neuron scale. For each $\theta_{i,j}^{l+1}$, $\phi(\varepsilon_{i,t}^l)$ produces a spike of 1 at exactly the inference moment of $t = l$, which equals to triggering $\theta_{i,j}^{l+1}$ to be updated at $t = l$. In this way, we can let the inference and weights update run all the time with $\phi(\cdot)$ modulating the local learning rate with

Table 2: Models and their properties.

| | Equivalence to BP | Local plasticity | Partial autonomy | Full autonomy |
|---|---|---|---|---|
| BP | ✓ | ✗ | ✗ | ✗ |
| IL | ✗ | ✓ | ✓ | ✗ |
| Z-IL | ✓ | ✓ | ✓ | ✗ |
| Fa-Z-IL | ✓ | ✓ | ✓ | ✓ |

Table 3: Success rate of detecting inference moments $t = l$ with $\phi(\cdot)$ of different $t_d$.

| $t_d$ | 1 | 2 | 3 | 4 | 5 | 6 | 7 | 8 | 16 |
|---|---|---|---|---|---|---|---|---|---|
| Success rate | 93.4% | 92.4% | 99.6% | 100.0% | 100.0% | 100.0% | 100.0% | 100.0% | 100.0% |

local information; thus, the resulting model works fully autonomously, performs all computations locally, and updates weights at $t = l$, i.e., producing exactly Z-IL/BP, as long as C1 and C3 are satisfied. Fa-Z-IL is summarized in Algorithm 2.

As the core of Fa-Z-IL, we found that quite simple functions $\phi(\cdot)$ can detect the inference moment of $t = l$ from $\varepsilon_{i,t}^l$. Specifically, from Lemma A.3 in the supplementary material, we know that $\varepsilon_{i,t}^l$ under C1 diverges from its stable states at exactly $t = l$, i.e., $\bar{\varepsilon}_{t<l}^l = \bar{0}$. Thus, $\phi(\cdot)$ can take in $\varepsilon_{i,t}^l$ and return 1 only if $\varepsilon_{i,t-t_d}^l = 0, \varepsilon_{i,t-t_d+1}^l = 0, \ldots, \varepsilon_{i,t-1}^l = 0, \varepsilon_{i,t}^l \neq 0$, where $t_d$ is a hyperparameter. In this way, $\phi(\varepsilon_{i,t}^l)$ detects the inference moment of $t = l$ and produces a spike of 1 at exactly $t = l$. In special cases where $\varepsilon_{i,t=l}^l = 0$, the detection fails, however, by Eq. (9), $\Delta\theta_{i,j}^{l+1}$ produced at $t = l$ is also zero, since $\varepsilon_{i,t=l}^l = 0$, so the failure does no harm. Since it is possible for $\varepsilon_{i,t}^l$ to go across zero, having a larger $t_d$ means a more accurate detection. In experiments, $\phi(\cdot)$ of $t_d = 4$ is already capable of detecting with 100% success rate; see Table 3. Furthermore, the function $\phi$, as the only additional component introduced in Fa-Z-IL to realize full autonomy, is highly plausible that such a computation could be performed by biological neurons, because some types of neurons are well-known to respond predominantly to changes in their input [63].

However, Fa-Z-IL does require the input to be presented before the teacher to satisfy C1, i.e., the convergence of a prediction phase before $t = 0$ is still needed. We consider this to be a requirement of the learning setup. Also, such a requirement is much weaker, compared to switching computational rules (BP) and detecting the convergence of global variables (IL). We leave the study of removing this requirement or putting it inside an autonomous neural system as future research. During this prediction phase before $t = 0$, it is notable that the error nodes change due to feedforward input, while during learning, the error nodes change due to feedback input. In order to prevent learning during prediction, $\phi$ is equal to 1 only if the change in error nodes is caused by feedback input. Furthermore, experiments of classification with Fa-Z-IL have been conducted, and Fa-Z-IL produces exactly the same result as Z-IL and BP, the numbers of which can be found in the supplementary material. It should also be noted that Fa-Z-IL loses formal equivalence to BP, but with $t_d > 4$, empirical equivalence always remains.

All proposed models are summarized in Table 2 (schematic algorithms are given in the supplementary material), where Fa-Z-IL is the only model that is not only equivalent to BP, but also performs all computations *locally* and *fully autonomously*.

## 5 Experiments

In this section, we complete the picture of this work with experimental results, providing ablation studies on MNIST and measuring the running time of the discussed approaches on ImageNet.

### 5.1 MNIST

We now show that zero-divergence cannot be achieved when strictly weaker conditions than C1, C2, and C3 in Section 3 are satisfied only. Specifically, with experiments on MNIST, we show the divergence of PCNs trained with ablated variants of Z-IL from ANNs trained with BP.

**Setup.** We use the same setup as in [48]. Specifically, we train for 64 epochs with $\alpha = 0.001$, batch size 20, and logistic sigmoid as $f$. To remove the stochasticity of the batch sampler, models in one group within which we measure divergence are set with the same seed. We evaluate three of such groups, each set with different randomly generated seeds ({1482555873, 698841058, 2283198659}), and the final numbers are averaged over them. The divergence is measured in terms of the error percentage on the test set summed over all epochs (test error, for short), as in [48, 37, 58, 53, 59], and of the final weights, as in [59]. The divergence of the test error is the L1 distance between the corresponding test errors, averaged over 64 training iterations (the test error is evaluated after each training iteration). The divergence of the final weights is the sum of the L2 distance between the

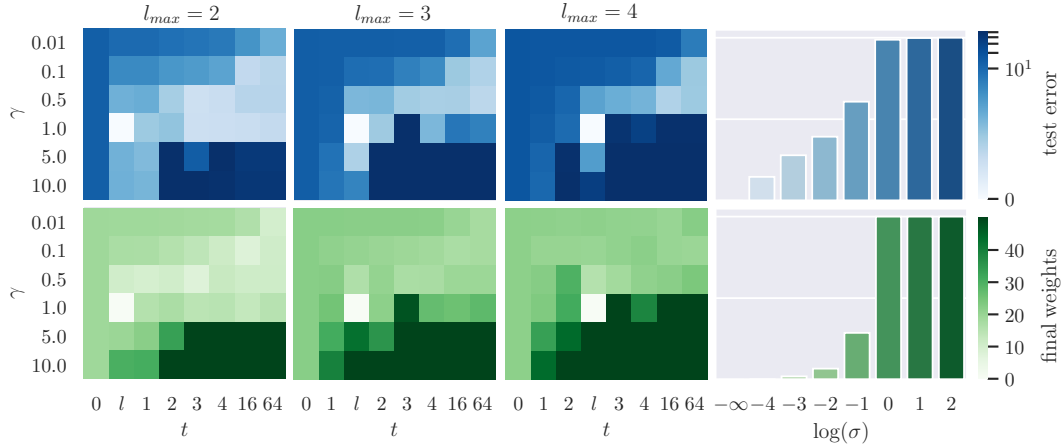

Figure 3: Ablation of C2, C3, and C1, where divergence is measured on test error and final weights.

corresponding weights, after the last training iteration. We conducted experiments on MNIST (with 784 input and 10 output neurons; the other settings are as in [64]). We investigated three different network structures with 1, 2, and 3 hidden layers, respectively (i.e., $l_{\max} \in \{2, 3, 4\}$), each containing 32 neurons.

**Ablation of C1.** Assuming C2 and C3 satisfied, to ablate C1, we consider situations when the network has not converged in the prediction stage before the training pair is presented, i.e., $x^l_{i,0} \neq \mu^l_{i,0}$. To simulate this, we sampled $x^l_{i,0}$ around the mean of $\mu^l_{i,0}$ with different standard deviations $\sigma$, where $\sigma = 0$ corresponds to satisfying C1. We swipe $\sigma = \{0, 0.0001, 0.001, 0.01, 0.1, 1, 10, 100\}$. Fig. 3, right column, shows that zero divergence is only achieved when $\sigma = 0$, i.e., C1 is satisfied. A larger version of Fig. 3 is given in the supplementary material.

**Ablation of C2 and C3.** Assuming C1 satisfied, to ablate C3, we swipe $\gamma = \{0.01, 0.1, 0.5, 1, 5, 10\}$, and to ablate C2, we swipe $t = \{l, 0, 1, 2, 3, 4, 16, 64\}$. Here, setting $t$ to a fixed number is exactly the implementation of IL with $t_c$ set to this fixed number. We set $t = l$ between $t = l_{\max} - 2$ and $t = l_{\max} - 1$ (as argued in the supplementary material). Fig. 3, three left columns, shows that zero divergence is only achieved when $t = l$ and $\gamma = 1$, i.e., C2 and C3 are satisfied. Note that the settings of $t \geq 16$ and $\gamma \leq 0.5$ are typical for IL in [20].

## 5.2 ImageNet

We further conduct experiments on ImageNet to measure the running time of Z-IL and Fa-Z-IL on large datasets. In detail, we show that Z-IL and Fa-Z-IL create minor overheads over BP, which supports their scalability. The detailed implementation setup is given in the supplementary material.

Table 4 shows the averaged running time of each weights update of BP, IL, Z-IL, and Fa-Z-IL. For IL, we set $t_c = 20$, following [48]. As can be seen, IL introduces large overheads, due to the fact that it needs at least $t_d$ inference steps before conducting a weights update. In contrast, Z-IL and Fa-Z-IL run with minor overheads compared to BP, as they require at most $l_{\max}$ inference steps to complete one update of weights in all layers. Comparing Fa-Z-IL to Z-IL, it is also obvious that the function $\phi$ creates only minor overheads. These observations support the claim that Z-IL and Fa-Z-IL are indeed scalable.

Table 4: Average runtime of each weights update (in ms) of BP, IL, Z-IL, and Fa-Z-IL.

| Devices | BP | IL | Z-IL | Fa-Z-IL |
|---------|-----|------|------|---------|
| CPU | 3.1 | 19.2 | 3.6 | 3.6 |
| GPU | 3.7 | 56.3 | 4.1 | 4.2 |

## 6 Related Work

We now review several lines of research that approximate BP in ANNs with BL models, each of which corresponds to an argument for which BP in ANNs is considered to be biologically implausible.

One such argument is that BP in ANNs encodes error terms in a biologically implausible way, i.e., error terms are not encoded locally. It is often discussed along with the lack of local plasticity. How error terms can be alternatively encoded and propagated locally has been one of the most intensively studied topics. One promising assumption is that the error term can be represented in dendrites of the corresponding neurons [35, 65, 66]. Such efforts are unified in [21], with a broad range of works [36, 37] encoding the error term in activity differences. IL and Z-IL in PCNs also do so. In addition, Z-IL shows that the error term in BP in ANNs can be encoded with its exact value in the prediction error of IL in PCNs at very specific inference moments.

BP in ANNs is also criticized for requiring an external control (e.g., the computation is changed between prediction and learning). An important property of PCNs trained with IL, in contrast, is that they autonomously [48] switch between prediction and learning. PCNs trained with Z-IL also have this property. But IL requires an external control over when to update weights ($t = t_c$), and Z-IL requires control to only update weights at very specific inference moments ($t = l$). However, Z-IL can be realized with full autonomy, leading to the proposed Fa-Z-IL.

Finally, BP in ANNs is also criticized for backward connections that are symmetric to forward connections in adjacent layers and for using unrealistic models of (non-spiking) neurons. A common way to remove the backward connections [8, 39, 67] is based on the idea of zeroth-order optimization [68]. However, the latter needs many trials depending on the number of weights [69] and can be further improved by perturbing the outputs of the neurons instead of the weights [70]. Admitting the existence of backward connections, more recent works show that asymmetric connections are able to produce a comparable performance [38, 71, 54]. In the predictive coding models (IL, Z-IL, and Fa-Z-IL), the errors are backpropagated by correct weights, because the model includes feedback connections that also learn. The weight modification rules for corresponding feedforward and feedback weights are the same, which ensures that they remain equal if initialized to equal values (see [48]). As for the models of neurons, BP has recently also been generalized to spiking neurons [72]. Our work is orthogonal to the above, i.e., we still use symmetric backward connections and do not consider spiking neurons.

Furthermore, [46] also analyzes the first steps during inference after a perturbation caused by turning on the teacher, starting from a prediction fixpoint, which may serve similar general purposes. However, the obtained learning rule differs substantially from that of our Z-IL, since the energy function is not the one which governs PCNs. A section outlining the differences of learning rules between [46] and Z-IL is included in the supplementary material.

Some features of PCNs are inconsistent with known properties of biological networks, but it has recently been shown that variants of PCNs can be developed without three of these implausible elements, which retain high classification performance. The first unrealistic feature are 1-to-1 connections between value and error nodes (Fig. 1). However, it has been proposed that errors can be represented in apical dendrites of cortical neurons [62], and the equations of predictive coding networks can be mapped on such an architecture [10]. Alternatively, it has been demonstrated how these 1-to-1 connections can be replaced by dense connections [73]. The other two unrealistic features of PCNs are symmetric forward-backward weights and non-linear functions affecting only some outputs of the neurons. Nevertheless, it has been demonstrated that these features may be removed without significantly affecting the classification performance [73].

# 7  Summary and Outlook

In this paper, we have presented for the first time a framework to BL that (1) produces exactly the same updates of the neural weights as BP, while (2) employing local plasticity. Based on the above framework, we have additionally presented a BL model that (3) also works fully autonomously. This suggests a positive answer to the long-disputed question whether the brain can perform BP.

As for future work, we believe that the proposed Fa-Z-IL model will open up many new research directions. Specifically, it has a very simple implementation (see Algorithm 2), with all computations being performed locally, simultaneously, and fully autonomously, which may lead to new architectures of neuromorphic computing.

## Broader Impact

This work shows that backpropagation in artificial neural networks can be implemented in a biologically plausible way, providing previously missing evidence to the debate on whether backpropagation could describe learning in the brain. In machine learning, backpropagation drives the contemporary flourish of machine intelligence. However, it has been doubted for long that though backpropagation is indeed powerful, its computational procedure is not possible to be implemented in the brain. Our work provides strong evidence that backpropagation can be implemented in the brain, which will solidify the community's confidence on pushing forward backpropagation-based machine intelligence. Specifically, the machine learning community may further explore if such equivalence holds for other or more complex BP-based networks.

In neuroscience, models based on backpropagation have helped to understand how information is processed in the visual system [15, 16]. However, it was not possible to fully rely on these insights, as backpropagation was so far seen unrealistic for the brain to implement. Our work provides strong confidence to remove such concerns, and thus could lead to a series of future works on understanding the brain with backpropagation. Specifically, the neuroscience community may now use the patterns produced by BP to verify if such computational model can explain learning in brain. Also, our work may inspire researchers to look for the existence of the function $\phi$, which completes the biological foundation of Fa-Z-IL.

As for ethical aspects and future societal consequences, we consider our work to be an important step towards understanding biological intelligence, which indicates at least no harm on the ethical aspects. Instead, being able to understand biological intelligence potentially leads to advances in medical research, which will substantially benefit the well-beings of humans.

## Acknowledgments and Disclosure of Funding

This work was supported by the China Scholarship Council under the State Scholarship Fund, by the National Natural Science Foundation of China under the grant 61906063, by the Natural Science Foundation of Tianjin City, China, under the grant 19JCQNJC00400, by the "100 Talents Plan" of Hebei Province, China, under the grant E2019050017, and by the Medical Research Council UK grant MC_UU_00003/1. This work was also supported by the Alan Turing Institute under the EPSRC grant EP/N510129/1 and by the AXA Research Fund.

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
