[Supplementary Material]

# Can the Brain Do Backpropagation?
# — Exact Implementation of Backpropagation in Predictive Coding Networks

**Yuhang Song**[1]**, Thomas Lukasiewicz**[1]**, Zhenghua Xu**[2,*]**, Rafal Bogacz**[3]
[1]Department of Computer Science, University of Oxford, UK
[2]State Key Laboratory of Reliability and Intelligence of Electrical Equipment,
Hebei University of Technology, Tianjin, China
[3]MRC Brain Network Dynamics Unit, University of Oxford, UK
yuhang.song@some.ox.ac.uk, thomas.lukasiewicz@cs.ox.ac.uk,
zhenghua.xu@hebut.edu.cn, rafal.bogacz@ndcn.ox.ac.uk

## A  Supplementary Material

### A.1  Bias Values as Parameters

We now add bias values as parameters, denoted by $v_i^{l+1}$ and $\beta_i^{l+1}$ for ANNs and PCNs, respectively.

Formally, in ANNs trained with BP, originally recalled in Section 2.1, Eq. (1) becomes:

$$y_i^l = \sum_{j=1}^{n^{l+1}} w_{i,j}^{l+1} f(y_j^{l+1}) + v_i^{l+1}. \tag{10}$$

Accordingly, we have an update rule for bias parameters (bias update) in addition to the one of the weight parameters (weight update):

$$\Delta v_{i,j}^{l+1} = -\alpha \cdot \partial E / \partial v_{i,j}^{l+1} = \alpha \cdot \delta_i^l. \tag{11}$$

Similarly, in PCNs trained with IL, originally recalled in Section 2.2, Eq. (5) becomes:

$$\mu_{i,t}^l = \sum_{j=1}^{n^{l+1}} \theta_{i,j}^{l+1} f(x_{j,t}^{l+1}) + \beta_i^{l+1} \quad \text{and} \quad \varepsilon_{i,t}^l = x_{i,t}^l - \mu_{i,t}^l. \tag{12}$$

Accordingly, we have an update rule for bias parameters in addition to the one of the weight parameters:

$$\Delta \beta_{i,j}^{l+1} = -\alpha \cdot \partial F_t / \partial \beta_{i,j}^{l+1} = \alpha \cdot \varepsilon_{i,t}^l. \tag{13}$$

Otherwise, all equations and conclusions still hold in Section 2.

In Section 3, we only need to add the assumption that the bias parameter for both ANNs and PCNs are also identical initially. To prove the conclusion of zero divergence of the weights update, the procedure remains unchanged. To prove the conclusion of zero divergence of the bias update, we only need Theorem 3.1, as it directly leads to the equivalence of bias update by Eqs. (11) and (13).

### A.2  Derivations of Eq. (7)

Before we start, we expand Eq. (6) with the definition of $\varepsilon_{i,t}^l = x_{i,t}^l - \mu_{i,t}^l$:

$$F_t = \sum_{l=0}^{l_{\max}-1} \sum_{i=1}^{n^l} \tfrac{1}{2} (\varepsilon_{i,t}^l)^2 = \sum_{l=0}^{l_{\max}-1} \sum_{i=1}^{n^l} \tfrac{1}{2} (x_{i,t}^l - \mu_{i,t}^l)^2. \tag{14}$$

Inference minimizes $F_t$ by modifying $x_{i,t}^l$ proportionally to the gradient of the objective function $F_t$. To calculate the derivative of $F_t$ over $x_{i,t}^l$, we note that each $x_{i,t}^l$ influences $F_t$ in two ways: (1) it

occurs in Eq. (14) explicitly, but (2) it also determines the values of $\mu_{k,t}^{l-1}$ via Eq. (5). Thus, the derivative contains two terms:

$$\Delta x_{i,t}^l = -\gamma \cdot \frac{\partial F_t}{\partial x_{i,t}^l} \tag{15}$$

$$= -\gamma \cdot \left( \frac{\partial \frac{1}{2}(x_{i,t}^l - \mu_{i,t}^l)^2}{\partial x_{i,t}^l} + \frac{\partial \sum_{k=1}^{n^{l-1}} \frac{1}{2}(x_{k,t}^{l-1} - \mu_{k,t}^{l-1})^2}{\partial x_{i,t}^l} \right) \tag{16}$$

$$= \gamma \cdot \left( -(x_{i,t}^l - \mu_{i,t}^l) + f'(x_{i,t}^l) \sum_{k=1}^{n^{l-1}} (x_{k,t}^{l-1} - \mu_{k,t}^{l-1})\theta_{k,i}^l \right) \tag{17}$$

$$= \gamma \cdot \left( -\varepsilon_{i,t}^l + f'(x_{i,t}^l) \sum_{k=1}^{n^{l-1}} \varepsilon_{k,t}^{l-1} \theta_{k,i}^l \right). \tag{18}$$

Considering also the special cases at $l = l_{\max}$ and $l = 0$, we obtain Eq. (7).

### A.3 Derivations of Eq. (9)

The weights update minimizes $F_t$ by modifying $\theta_{i,j}^{l+1}$ proportionally to the gradient of the objective function $F_t$. To compute the derivative of the objective function $F_t$ over $\theta_{i,j}^{l+1}$, we note that $\theta_{i,j}^{l+1}$ affects the value of the function $F_t$ of Eq. (14) by influencing $\mu_{i,t}^l$ via Eq. (5), hence,

$$\Delta \theta_{i,j}^{l+1} = -\alpha \cdot \partial F_t / \partial \theta_{i,j}^{l+1} \tag{19}$$

$$= -\alpha \cdot \frac{\partial \frac{1}{2}(x_{i,t}^l - \mu_{i,t}^l)^2}{\partial \theta_{i,j}^{l+1}} \tag{20}$$

$$= \alpha \cdot \varepsilon_{i,t}^l f(x_{j,t}^{l+1}). \tag{21}$$

### A.4 Proof of Theorems 3.1 and 3.2

**Theorem 3.1.** *Let $M$ be a PCN, $M'$ be its corresponding ANN (with the same initial weights as $M$), and let $s$ be a datapoint. Then, every prediction error $\varepsilon_{i,t}^l$ at $t = l$, $l \in \{0, \dots, l_{max}-1\}$, in $M$ trained with IL on $s$ under C1 and C3 is equal to the error term $\delta_i^l$ in $M'$ trained with BP on $s$.*

*Proof.* We first note that $\varepsilon_{i,t}^l$ under C2 is $\varepsilon_{i,l}^l$. We give a proof by induction on the depth $l$ of the PCNs and ANNs. For PCNs, as $t = l$, it is also inducing on the inference moments.

♢ *Base Case*: If $l = 0$,

putting C1 $\mu_{i,0}^l = y_i^l$ into Eq. (8), and by comparison with the first case in Eq. (4): $\varepsilon_{i,l}^l = \delta_i^l$.

♢ *Induction Step*: For $l \in \{1, \dots, l_{\max} - 1\}$,

$$\varepsilon_{i,l}^l = f'(\mu_{i,0}^l) \sum_{k=1}^{n^{l-1}} \varepsilon_{k,l-1}^{l-1} \theta_{k,i}^l, \text{ by Lemma A.4;}$$

$$\delta_i^l = f'(y_i^l) \sum_{k=1}^{n^{l-1}} \delta_k^{l-1} w_{k,i}^l, \text{ by the second case in Eq. (4);}$$

$$\mu_{i,0}^l = y_i^l, \text{ by C1;}$$

$$w_{i,j}^l = \theta_{i,j}^l, \text{ as corresponding initial weights in both models are assumed to be identical;}$$

$$\varepsilon_{i,l}^l = \delta_i^l, \text{ if } \varepsilon_{k,l-1}^{l-1} = \delta_k^{l-1}.$$

□

**Theorem 3.2.** *Let $M$ be a PCN, $M'$ be its corresponding ANN (with the same initial weights as $M$), and let $s$ be a datapoint. Then, every update $\Delta\theta_{i,j}^{l+1}$ at $t = l$, $l \in \{0, \dots, l_{max}-1\}$, in $M$ trained with IL on $s$ under C1 and C3 is equal to the update $\Delta w_{i,j}^{l+1}$ in $M'$ trained with BP on $s$.*

*Proof.* Looking at how $\partial F_t / \partial \theta_{i,j}^{l+1}$ and $\partial E / \partial w_{i,j}^{l+1}$ can be computed via Eqs. (9) and (3), and putting C2 $t = l$ into the equation,

$$\Delta \theta_{i,j}^{l+1} = \alpha \cdot \varepsilon_{i,l}^l f(x_{j,l}^{l+1}), \tag{22}$$

$$\Delta w_{i,j}^{l+1} = \alpha \cdot \delta_i^l f(y_j^{l+1}). \tag{23}$$

We notice that one of the terms in both equations are equivalent according to Theorem 3.1: $\varepsilon_{i,l}^l = \delta_i^l$. Thus, in the following, we focus on proving that the other terms in both equations are identical: $f(x_{j,l}^{l+1}) = f(y_j^{l+1})$. First, C1 provides the base to start, which is the equivalence of the initial state between IL and BP under C1, C2, and C3: $x_{j,0}^{l+1} = \mu_{j,0}^{l+1} = y_j^{l+1}$. Then, Lemma A.3 links later inference moments of IL under C1, C2, and C3 to its initial state: $x_{j,l}^{l+1} = x_{j,0}^{l+1}$. Thus, we have $f(x_{j,l}^{l+1}) = f(y_j^{l+1})$. $\qquad \square$

**Lemma A.3.** *Under C1, so that a variable at a specific layer may diverge from its corresponding initial stable states, it needs specific inference steps related to the layer that the variable belongs to. Formally,*

$$\overline{x}_{t<l}^l = \overline{x}_0^l, \overline{\varepsilon}_{t<l}^l = \overline{\varepsilon}_0^l = 0, \overline{\mu}_{t<l}^{l-1} = \overline{\mu}_0^{l-1}, \text{ for } l \in \{1, \ldots, l_{max} - 1\}, \text{ i.e.,}$$
$$\Delta \overline{x}_{t<l-1}^l = \overline{0}, \Delta \overline{\varepsilon}_{t<l-1}^l = \overline{0}, \Delta \overline{\mu}_{t<l-1}^{l-1} = \overline{0}, \text{ for } l \in \{1, \ldots, l_{max} - 1\}.$$

*Proof.* Starting from the inference moment $t = 0$, $\overline{x}_0^0$ is dragged away from $\overline{\mu}_0^0$ and fixed to $\overline{s}^{\text{out}}$, i.e., $\overline{\varepsilon}_0^0$ turns into nonzero from zero. Since $\overline{x}$ in each layer is updated only on the basis of $\overline{\varepsilon}$ in the same and previous adjacent layer, as indicated by Eq. (7), also considering C1, $\overline{\varepsilon}$ is initially zero for all layers but the output layer, it will take $l$ time steps to modify $\overline{x}_t^l$ at layer $l$ from the initial state. Hence, $\overline{x}_t^l$ will remain in that initial state $\overline{x}_0^l$ for all $t < l$, i.e., $\overline{x}_{t<l}^l = \overline{x}_0^l$. Furthermore, any change in $\overline{x}_t^l$ causes a change in $\overline{\varepsilon}_t^l$ and $\overline{\mu}_t^{l-1}$ instantly via Eq. (5) (otherwise $\overline{\varepsilon}_t^l$ and $\overline{\mu}_t^{l-1}$ remain in their corresponding initial states). Thus, we know $\overline{\varepsilon}_{t<l}^l = \overline{\varepsilon}_0^l$ and $\overline{\mu}_{t<l}^{l-1} = \overline{\mu}_0^{l-1}$. Also, according to C1, $\overline{\varepsilon}_{t<l}^l = \overline{\varepsilon}_0^l = 0$. Equivalently, we have $\Delta \overline{x}_{t<l-1}^l = \overline{0}$, $\Delta \overline{\varepsilon}_{t<l-1}^l = \overline{0}$, and $\Delta \overline{\mu}_{t<l-1}^{l-1} = \overline{0}$. $\qquad \square$

**Lemma A.4.** *The prediction error of IL $\varepsilon_{i,t}^l$ at $t = l$ (i.e., $\varepsilon_{i,l}^l$) under C1 and C3 can be derived from itself at previous inference moments in the previous layer: $\varepsilon_{k,t}^{l-1}$ at $t = l - 1$ (i.e., $\varepsilon_{i,l-1}^{l-1}$). Formally,*

$$\varepsilon_{i,l}^l = f'(\mu_{i,0}^l) \sum_{k=1}^{n^{l-1}} \varepsilon_{k,l-1}^{l-1} \theta_{k,i}^l, \text{ for } l \in \{1, \ldots, l_{max} - 1\}. \tag{24}$$

*Proof.* We first write a dynamic version of $\varepsilon_{i,t}^l = x_{i,t}^l - \mu_{i,t}^l$:

$$\varepsilon_{i,t}^l = \varepsilon_{i,t-1}^l + \left(\Delta x_{i,t-1}^l - \Delta \mu_{i,t-1}^l\right), \tag{25}$$

where $\Delta \mu_{i,t-1}^l = \mu_{i,t}^l - \mu_{i,t-1}^l$. Then, we expand $\varepsilon_{i,l}^l$ with the above equation and simplify it with Lemma A.3, i.e., $\varepsilon_{i,t<l}^l = 0$ and $\Delta \mu_{i,t<l-1}^{l-1} = 0$:

$$\varepsilon_{i,l}^l = \varepsilon_{i,l-1}^l + \left(\Delta x_{i,l-1}^l - \Delta \mu_{i,l-1}^l\right) = \Delta x_{i,l-1}^l, \text{ for } l \in \{1, \ldots, l_{\max} - 1\}. \tag{26}$$

We further investigate $\Delta x_{i,l-1}^l$ expanded with the inference dynamic Eq. (7) and simplify it with Lemma A.3, i.e., $\varepsilon_{i,t<l}^l = 0$,

$$\Delta x_{i,l-1}^l = \gamma(-\varepsilon_{i,l-1}^l + f'(x_{i,l-1}^l)) \sum_{k=1}^{n^{l-1}} \varepsilon_{k,l-1}^{l-1} \theta_{k,i}^l = \gamma f'(x_{i,l-1}^l) \sum_{k=1}^{n^{l-1}} \varepsilon_{k,l-1}^{l-1} \theta_{k,i}^l, \text{ for } l \in \{1, \ldots, l_{\max} - 1\}. \tag{27}$$

Putting Eq. (27) into Eq. (26), we obtain:

$$\varepsilon_{i,l}^l = \gamma f'(x_{i,l-1}^l) \sum_{k=1}^{n^{l-1}} \varepsilon_{k,l-1}^{l-1} \theta_{k,i}^l, \text{ for } l \in \{1, \ldots, l_{\max} - 1\}. \tag{28}$$

With Lemma A.3, $x_{i,l-1}^l$ can be replaced with $x_{i,0}^l$. With C1, we can further replace $x_{i,0}^l$ with $\mu_{i,0}^l$. Thus, the above equation becomes:

$$\varepsilon_{i,l}^l = \gamma f'(\mu_{i,0}^l) \sum_{k=1}^{n^{l-1}} \varepsilon_{k,l-1}^{l-1} \theta_{k,i}^l, \text{ for } l \in \{1, \ldots, l_{\max} - 1\}. \tag{29}$$

Then, put C3, $\gamma = 1$, into the above equation. $\qquad \square$

## A.5 Solely Relaxing C3

Solely relaxing C3 will result in the conclusion of Lemma A.4, i.e., Eq. (24) changing to:

$$\varepsilon_{i,l}^l = \gamma f'(\mu_{i,0}^l)\sum_{k=1}^{n^{l-1}} \varepsilon_{k,l-1}^{l-1}\theta_{k,i}^l, \text{ for } l \in \{1,\ldots,l_{\max}-1\}, \tag{30}$$

since the derivation of Lemma A.4 terminates at Eq. (29). It further causes the conclusion of Theorem 3.1 changing from $\varepsilon_{i,t}^l = \delta_i^l$ to $\varepsilon_{i,t}^l = \gamma^l \delta_i^l$ at $t = l$, the proof of which is the same as that of the original Theorem 3.1 but using the new Lemma A.4. The change in the conclusion of Theorem 3.1 immediately changes the conclusion of Theorem 3.2 from $\partial F_t/\partial\theta_{i,j}^{l+1} = \partial E/\partial w_{i,j}^{l+1}$ to $\partial F_t/\partial\theta_{i,j}^{l+1} = \gamma^l \partial E/\partial w_{i,j}^{l+1}$, where $t = l$. Thus, solely relaxing the condition $\gamma = 1$, i.e., relaxing C3 while keeping C1 and C2 satisfied, results in BP with a different learning rate for different layers, where $\gamma$ is the decay factor of this learning rate along layers.

## A.6 Schematic Algorithms

---
**Algorithm 1** Learning one training pair $(\overline{s}^{\text{in}}, \overline{s}^{\text{out}})$ (presented for the duration $T$) with IL
---
**Require:** $\overline{x}_0^{l_{\max}}$ is fixed to $\overline{s}^{\text{in}}$, $\overline{x}_0^0$ is fixed to $\overline{s}^{\text{out}}$.
1: **for** $t = 0$ to $T$ **do**      // presenting a training pair
2:    **for** each neuron $i$ in each level $l$ **do**      // in parallel in the brain
3:      Update $x_{i,t}^l$ to minimize $F_t$ via Eq. (7)      // inference
4:      **if** $t = t_c$ **then**      // external control signal
5:        Update each $\theta_{i,j}^{l+1}$ to minimize $F_t$ via Eq. (9)
6:        **return**      // the brain rests
7:      **end if**
8:    **end for**
9: **end for**
---

---
**Algorithm 2** Learning one training pair $(\overline{s}^{\text{in}}, \overline{s}^{\text{out}})$ (presented for the duration $T$) with Z-IL
---
**Require:** $\overline{x}_0^{l_{\max}}$ is fixed to $\overline{s}^{\text{in}}$, $\overline{x}_0^0$ is fixed to $\overline{s}^{\text{out}}$.
**Require:** $x_{i,0}^l = \mu_{i,0}^l$ for $l \in \{1,\ldots,l_{\max}-1\}$ (C1), and $\gamma = 1$ (C3).
1: **for** $t = 0$ to $T$ **do**      // presenting a training pair
2:    **for** each neuron $i$ in each level $l$ **do**      // in parallel in the brain
3:      Update $x_{i,t}^l$ to minimize $F_t$ via Eq. (7)      // inference
4:      **if** $t = l$ **then**      // external control signal
5:        Update each $\theta_{i,j}^{l+1}$ to minimize $F_t$ via Eq. (9)
6:      **end if**
7:      **if** $t = l_{\max}$ **then**
8:        **return**      // the brain rests
9:      **end if**
10:    **end for**
11: **end for**
---

---
**Algorithm 3** Learning one training pair $(\overline{s}^{\text{in}}, \overline{s}^{\text{out}})$ (presented for the duration $T$) with Fa-Z-IL
---
**Require:** $\overline{x}_0^{l_{\max}}$ is fixed to $\overline{s}^{\text{in}}$, $\overline{x}_0^0$ is fixed to $\overline{s}^{\text{out}}$.
**Require:** $x_{i,0}^l = \mu_{i,0}^l$ for $l \in \{1,\ldots,l_{\max}-1\}$ (C1), and $\gamma = 1$ (C3).
1: **for** $t = 0$ to $T$ **do**      // presenting a training pair
2:    **for** each neuron $i$ in each level $l$ **do**      // in parallel in the brain
3:      Update $x_{i,t}^l$ to minimize $F_t$ via Eq. (7)      // inference
4:      Update each $\theta_{i,j}^{l+1}$ to minimize $F_t$ via Eq. (9) with learning rate $\alpha \cdot \phi(\varepsilon_{i,t}^l)$
5:    **end for**
6: **end for**
---

Schematic algorithms of IL, Z-IL, and Fa-Z-IL are provided in Algorithms 1, 2, and 3 for comparison, respectively, the two of which, IL and Fa-Z-IL are already provided in the paper.

Table 1: Unscaled data in Fig. 2.

| $\sigma$ | 0 | 0.0001 | 0.001 | 0.01 | 0.1 | 1 | 10 | 100 |
|---|---|---|---|---|---|---|---|---|
| Divergence of test error | **0** | $4.23\times10^{-3}$ | $1.77\times10^{-2}$ | $6.05\times10^{-2}$ | $6.15\times10^{-1}$ | $3.73\times10^{1}$ | $4.17\times10^{1}$ | $4.26\times10^{1}$ |
| Divergence of final weights | $\mathbf{2.37\times10^{-13}}$ | $2.40\times10^{-2}$ | $9.94\times10^{-1}$ | $3.36\times10^{0}$ | $1.43\times10^{1}$ | $1.08\times10^{2}$ | $1.12\times10^{3}$ | $1.22\times10^{4}$ |

Table 2: Test error of different $\sigma$.

| $\sigma$ | 0 | $10^{-4}$ | $10^{-3}$ | $10^{-2}$ | $10^{-1}$ | 1 | $10^{1}$ | $10^{2}$ |
|---|---|---|---|---|---|---|---|---|
| Test error | $1.0546\times10^{-1}$ | $1.0548\times10^{-1}$ | $1.0557\times10^{-1}$ | $1.0596\times10^{-1}$ | $1.1507\times10^{-1}$ | $6.8915\times10^{-1}$ | $7.5753\times10^{-1}$ | $7.7116\times10^{-1}$ |

## A.7   Large Figures

Larger versions (with numbers as annotations) of the subfigures in Fig. 3 are given in Figs. 1 and 2. Note that the divergence of final weights is not exactly zero but very small ($< 10^{-12}$), due to rounding errors.

## A.8   Why Put $t = l$ Between $t = l_{\max} - 2$ and $t = l_{\max} - 1$

We put Z-IL ($t = l$) between $t = l_{\max} - 2$ and $t = l_{\max} - 1$ for models with different $l_{\max}$. This is due to the fact that Z-IL ($t = l$) is doing inference at a degree between $t = l_{\max} - 2$ and $t = l_{\max} - 1$. Specifically, Z-IL ($t = l$) uses $F_{l_{\max}-1}$ to update $\theta_{i,j}^{l_{\max}}$, which is a result from doing inference at $t = l_{\max} - 2$, however, Z-IL did not use $F_{l_{\max}-1}$ for weights update in all layers, instead it uses $F_{l<l_{\max}-1}$ at previous inference moments to update weights $\theta_{i,j}^{l}$ in other layers $l < l_{\max}$. Thus, we consider that Z-IL ($t = l$) lies between $t = l_{\max} - 2$ and $t = l_{\max} - 1$ from the perspective how much inference is conducted. For example, for $l_{\max} = 3$, we present the result as $t = \{0, 1, l, 2, \ldots\}$.

## A.9   Detailed Setup of Experiments on ImageNet

Experiments are conducted on 2 GPUs of Nvidia GeForce GTX 1080Ti, and 8 CPUs of Intel Core i7, with 32 GB RAM. The batch size is set to 1 to test the running time. The input size of the image is $224 \times 224$ gray scale, we only implemented fully connected layers: the hidden layers are of size 4096, 2048, and 1024, respectively. The size of the output layer is 27, corresponding to the 27 high-level categories in ImageNet. All dependencies and their versions are specified as follows:

```
torchvision ==0.4.2
setproctitle ==1.1.10
psutil ==5.6.3
requests ==2.22.0
pillow ==6.1
pandas ==0.24.2
matplotlib ==3.1.0
seaborn ==0.9.0
numpy ==1.16.4
tensorflow ==1.15
pyinquirer ==1.0.3
torch ==1.3.1
ray ==0.7.4
pprint ==0.1
tabulate ==0.8.7
opencv-python ==4.2.0.34
PyInquirer ==1.0.3
```

## A.10   Test error of all situations

Note that in all our experiments, Fa-Z-IL produces the same test error as Z-IL, thus, no separated tables or figures for Fa-Z-IL are provided. Table 2 shows the test error of different $\sigma$.

(a) divergence of test error ($l_{\max} = 2$)

(b) divergence of final weights ($l_{\max} = 2$)

(c) divergence of test error ($l_{\max} = 3$)

(d) divergence of final weights ($l_{\max} = 3$)

(e) divergence of test error ($l_{\max} = 4$)

(f) divergence of final weights ($l_{\max} = 4$)

Figure 1: Larger versions (with numbers as annotations) of the subfigures in Fig. 3.

Figure 2: Larger versions (with numbers as annotations) of the subfigures in Fig. 3. For visualization, the divergence of test error has been normalized to $[0, 1]$ by a logarithmic scale and then clipped to $[0, 1]$; the divergence of the final weights has been normalized linearly to $[0, 1]$ from min of 0 and max of 50, values larger than 50 have been clipped to 50. The unscaled data are given in Table 1.

(a) $l_{\max} = 2$

(b) $l_{\max} = 3$

(c) $l_{\max} = 4$

Figure 3: Test error of different $t$ and $\gamma$.

## A.11 Differences of learning rules between [46] and Z-IL

The energy function of [46] is closely related to the Hopfield net (but with latent variables) or with the Boltzmann machine energy function (but with continuous nonlinearities inserted), best expressed with notations and setup in this paper as:

$$E_t = \sum_i \frac{1}{2}(s_i)^2 - \frac{1}{2}\sum_{i \neq j}\eta_{i,j}f(s_i)f(s_j) - \sum_i b_i f(s_i), \tag{31}$$

where the support of $s_i$ is given by all the value nodes in the neural network, $\eta_{i,j}$ is the synapse weight from node $j$ to node $i$, $b_i$ is the bias term, and $f$ is the nonlinear activation function. While the energy function of Z-IL is simply the energy of the error nodes:

$$F_t = \sum_{l=0}^{l_{\max}-1}\sum_{i=1}^{n^l}\frac{1}{2}(\varepsilon_{i,t}^l)^2 \,. \tag{32}$$

The other learning rules thus diverge significantly from there. We leave the discussion of how the two works could be related regardless of such huge differences as future research.