[Reviews · NeurIPS 2020]

Review 1

Summary and Contributions: This paper starts from the inference learning (IL) approach by Whittington and Bogacz for predictive coding networks (PCNs) and makes two new contributions. First, it is shown that under specific conditions, (zero-divergent) IL is exactly equivalent to backpropagation (BP). Second, an extension (Fa-Z-IL) is introduced which can autonomously switch between prediction and learning.

Strengths: The paper makes a new contribution to the recent focus in creating biologically plausible variants of backpropagation. It shows that PCN is equivalent to backpropagation under specific assumptions and do this is a fully autonomous manner while using relatively simple local update rules.

Weaknesses: It remains somewhat unclear if the introduced constraints (C1-C3) as well as the function phi to modulate local learning in the fully autonomous setting are truly biologically plausible. Other criticism for BP remain for the current approach. I.e. the weight transport problem, the question how to generalize to spiking neural networks and how to apply this recurrent neural networks without the need to transport information back through time.

Correctness: Claims and methods seem correct as well as the employed empirical methodology.

Clarity: yes.

Relation to Prior Work: Other recent work claims that predictive coding converges to the backpropagation gradients (https://arxiv.org/pdf/2006.04182.pdf). It is not discussed how those findings relate to the present work.

Reproducibility: Yes

Additional Feedback:


Review 2

Summary and Contributions: The authors propose a model for how backpropagation (BP) could be implemented in feed-forward networks of the brain. They claim that they solve three issues: (1) their model produces exaclty the same updates as BP, (2) it uses only local plasticity, and (3) it is fully autonomous. The paper is clearly biologically motivated. The model is based on previous work on predictive coding networks (PCNs). The connection btw. PCNs and backprop has been introduced previously [ref. 48 in the paper]. In fact the proposed model is a simple extension of this. Contributions - They proof that weight updates can be defined in a PCN that are mathematically equivalent to BP updates in a feed-forward network. - They claim that these update rules are local. I do not completely agree with this claim. - They show that the updates can be done fully autonomously without the need of external control (except for clamping of the output neurons to the targets). I am not sure whether this is the case (see point 4 below). In general, I think that the paper presents an interesting study. Some questions need to be clarified.

Strengths: The topic of the paper is interesting, although a bit over-researched in my opinion. I am personally not too enthusiastic about the idea that BP could be implemented in the brain, but I acknowledge that others are. The manuscript is well written and understandable. The paper includes an interesting theoretical grounding. They proof that weight updates can be defined in a PCN that are mathematically equivalent to BP updates in a feed-forward network.

Weaknesses: I have some critical remarks: 1.) Weight transport problem. This problem is not solved in the model. In fact the model needs symmetric weights. Feedback alignment will probably not work here, as I assume that the existence of an equilibrium state necessitates symmetric weights. 2.) Locality of learning rule. The authors claim that the update rules are local. Whether this is true however depends on the definition of locality. Let us look at a specific connection theta^{l+1}_{i,j}. This weight connects the neuron x^{l+1}_j to a summing node mu^l_i. The update however depends on x^l_i [correction: epsilon^l_i], a node which is neither pre-nor postsynaptic to the synapse. Therefore, the update rule is not local to the synapse. One could say that it is local if one assumes that x^l_i is spatially close to mu^l_i and then define locality as spatial proximity. But from the biological viewpoint, is there any evidence for such updates? 3.) Confusion about inference. In C1 at p.5 the authors say that the condition C1 is natrually satisfied if before start of training over a trainig pair, the network has converged in the prediction stage. This makes perfect sense to me. But then later on, in Alg 2 as well as in Fig. 2 and in the text, this initial convergence (for each training example) does not appear anymore, somehow suggestiong that this convergence phase is missing in the algorithm. Please clarify. 4.) Fully autonomous learning Given that the convergence discussed in (3) above is necessary, then I don't understand the autonomy using the described mechanism. The simple function phi outputs a spike when the error node changes from being 0 (for some time t_d) to something non-zero, which induces plasticity. However, this is also the case during the initial convergence. Please explain why this is not a problem.

Correctness: Yes, but see my remarks above.

Clarity: Yes, the paper is well-written, except for my point (3) above.

Relation to Prior Work: Yes. The authors deal with feed-forward neural networks. They do not discuss the case of recurrent networks, which is in my opinion the more relevant case when it comes to biology. The recurrent case has also been tackled and should be discussed briefly. See e.g. Murray, J. M. Local online learning in recurrent networks with random feedback. eLife (2019). Bellec, G. et al., A solution to the learning dilemma for recurrent networks of spiking neurons. bioRxiv (2019).

Reproducibility: Yes

Additional Feedback: see points above. After Author response: Locality of learning: I mean "The update however depends on epsilon^l_i, a node which is neither pre-nor postsynaptic to the synapse." Fully autonomous learning: The authors propose that Phi should only equal 1 if the change in the error node is caused by feedback input. I am not sure whether this works. It should be explained in detail. In general, I think that the equivalence with BP is interesting.


Review 3

Summary and Contributions: This paper derives a set of conditions under which the weight update in predictive coding networks (PCNs) becomes equivalent to error backpropagation (BP) on a corresponding artificial neural network (ANN). These conditions (Z-IL, which complement previous findings [48]) inspire the derivation of a more plausible algorithm, termed Fa-Z-IL. A small set of experiments is carried out, mainly to verify deviations from BP and the runtime of the algorithms. Update: ***** After reading the authors' response and discussing I have raised my score, but I would ask the authors to clarify the autonomy limitations that arise from C1.

Strengths: - The paper presents results for a framework of great relevance in neuroscience. - The approach is theoretically grounded. - Gating synaptic plasticity by the change in postsynaptic activity is an interesting idea.

Weaknesses: - The autonomy achieved by Fa-Z-IL seems limited: see first point under 'Correctness' below. - On plausibility: PCNs (at least in their present implementation) have a number of implausible aspects that are not acknowledged or sufficiently discussed in this paper, unlike in [48]. For example, errors are propagated backwards linearly (after multiplication/modulation by f'), and error neurons and prediction neurons connect with special 1-to-1 connections. Thus, it is not straightforward to design neuromorphic or more biologically-realistic spiking neuron implementations that are consistent with the model. - On the framing of the paper: I see the approach more as an attempt to speed-up the algorithm of [48] than that of granting it additional autonomy. Note that, as proposed in [48], one could introduce a plasticity trigger which depends on the rate of change. This requires (at least formally, at first sight) a full relaxation in the presence of a teacher, but it would make the model autonomous in the sense argued by the authors. - The experiments are fairly incomplete. Since Fa-Z-IL is no longer (in my understanding) exactly equivalent to BP it would be important to check whether one can actually train a model using the proposed \phi.

Correctness: Major clarification request to be addressed by the authors: - The paper claims that Fa-Z-IL is 'fully autonomous' (FA). However, the algorithm requires the network state to be initialized in accordance to C1. How can this be done fully autonomously? When the data point switches, the dynamics (7) will not immediately lead to the prediction fixed point. The natural way to fulfill C1 is to run the dynamics in a first phase without the teacher being present. Note that the error neurons will not necessarily be zero during this relaxation phase. How does Fa-Z-IL deal with this first initialization phase? Or do the authors have something else in mind that I missed? Furthermore: - I found the title too grandiose and disconnected from the actual presented approach. It could equally well apply to a larger number of the cited papers in the literature. - It should be more clearly stated that when going from Z-IL to Fa-Z-IL, strictly speaking one loses the formal equivalence to BP.

Clarity: Overall, the paper is clear and well-written. Just a minor suggestion on Sect. 2.1: I think it's fair to assume most of the readers are more familiar with the ANN layer numbering conventions. It could therefore be better to adapt the notation of PCNs to the more usual one (input < output layer indices).

Relation to Prior Work: The paper cites and discusses a large body of appropriate work. However: - The initialization stated here as C1 was already proposed (and in fact used) by [48]. This must be acknowledged. - It should also be clarified (for example, around line 99) that [48] already provided formal conditions under which IL and BP become formally equivalent. In particular [48] provides a condition on the precision of the output teacher variables. In practice, this results in a teacher that weakly perturbs the network, similarly to the requirement that arises in many of the cited learning algorithms which approximate BP. - It would be good to discuss in some more detail ref. [46]. There, the authors analyze the first steps during inference after a perturbation caused by turning on the teacher, starting from a prediction fixed point, like here. The obtained learning rule differs substantially since the energy function is not the one which governs PCNs.

Reproducibility: Yes

Additional Feedback:


Review 4

Summary and Contributions: This paper proposes a predictive coding network model which updates network weights that mimics backpropagation. Previous studies (such as [48]) have shown the relationship between PCN and BP, the main contribution of this study is the modification of the PCN learning algorithm so that it can produce the same weight updates as BP.

Strengths: The topic is interesting and the method is technically sound. It implements the BP algorithm in a biologically plausible way. Based on an existing predictive coding network [48], the main work of this study lies in that: 1) it extends inference learning (IL) in PCN with three conditions (Z-IL) to constrain that Z-IL and BP produces the same weight updates under the same initialization; 2) it implements an automatic version of the proposed approach (Fa-Z-IL).

Weaknesses: 1. The title of the paper seems to be over-claimed. PCN model can update weights in the way backpropagation does, but does it mean the brain can do backpropagation? 2. Some claims are too strong. For example, "this work fully bridges the crucial gaps between BP and BL, provides previously missing evidence to the debate on whether BP could describe learning in the brain, and links the power of biological and machine intelligence. "

Correctness: Mostly correct.

Clarity: The paper is well organized and easy to follow.

Relation to Prior Work: Yes.

Reproducibility: Yes

Additional Feedback: 1. In the MNIST experiment, ablation test is carried out to evaluate the effectiveness of the three conditions. What's the classification accuracy under different situations? 2. In Fig. 3, how were the test error and the final weights computed? The authors claim that "The divergence of the test error and the final weights is measured by the L1 and L2 distances, respectively." Detail of how these two criteria are computed is needed. 3. Figure 2 is difficult to understand. Extra description is required. Typo: Line 292: networs -> networks

[Author Response · NeurIPS 2020]

We thank all reviewers for their comments. We will correct all typos and address all minor comments in the final paper.

**Reviewer #1:** Biological plausibility of C1-C3 and $\phi$?　**C1:** Given an input-target pair, C1 means that a model
makes a prediction from the input, before it learns from the target, which is quite natural for biological neural systems.
**C2 and $\phi$:** C2 is realized by $\phi$, and it is very plausible that $\phi$ can be performed by biological neurons, because some
types of neurons are well-known to respond predominantly to changes in their input [71]. **C3:** Though it is unnatural
for biological neural systems to have a specific integration step, we show in the supplementary material that relaxing C3
results in BP with a different learning rate for different layers, which does not violate the core of BP.
Weight transport problem?　In the predictive coding model, the errors are back-propagated by correct weights, because
the model includes feedback connections that also learn. The weight modification rules for corresponding feedforward
and feedback weights are the same, which ensures that they remain equal if initialized to equal values (see [48]).
Other criticisms for BP (spiking and recurrent neural networks) remain?　As discussed in the related work section, they
are left as future work, while this work addresses two of the most crucial ones: local plasticity and autonomy.
Related work of https://arxiv.org/abs/2006.04182?　This work was published on 7 Jun 2020 (after the NeurIPS'20
deadline). It includes another approximation to BP but no equivalence; we will add it to the related work.

**Reviewer #2:** Weight transport problem?　Please see line 8 in the response to Reviewer #1.
Locality of learning rule? (the update of $\theta_{i,j}^{l+1}$ depends on $x_i^l$)?　According to Eq. 9, the update of $\theta_{i,j}^{l+1}$ actually depends
on $x_j^{l+1}$ rather than $x_i^l$ (we suspect the reviewer might have misread Eq. 9), so it is local.
Confusion about inference?　Alg. 2 explicitly states C1 in the second "Require". In line 158, we make a note that C1 is
omitted in Fig. 2 for simplicity. This note will be included in the caption of Fig. 2.
Full autonomy during the initial convergence of prediction?　During the prediction phase, the error nodes change due
to feedforward input, while during learning, the error nodes change due to feedback input. We will clarify that, in order
to prevent learning during prediction, $\phi$ is equal to 1 only if the change in error node is caused by feedback input.

**Reviewer #3:** Limitations of PCNs.　They are well-discussed in [10,48]; we will add a summary of them. Note that
some limitations have been addressed (e.g., 1-to-1 connections are addressed in [10]). We will also review key studies
illustrating that PCNs are widely used and informative models of information processing in the brain.
Model in [48] already autonomous, because plasticity triggered by network convergence?　The plasticity trigger in [48]
requires global information (i.e., total error in all error nodes), while our trigger $\phi$ needs only local information, thus, is
more plausible for biological implementation.
Experiment of Fa-Z-IL?　$\phi$ with $t_d > 4$ succeeds in all detections (Table 3), i.e., Fa-Z-IL with such $\phi$ coincides with
Z-IL (BP). We will include the classification results of such Fa-Z-IL, which produces the same results as Z-IL (BP).
Clarification of full autonomy?　Fa-Z-IL does require the input to be presented before the teacher to satisfy C1. We
consider this to be a requirement of the learning setup and will make it explicit in the paper, moderating our claim of
autonomy. However, we consider such requirement to be much weaker, compared to switching computational rules
(BP) and detecting convergence of global variables (IL). We leave the study of removing this requirement or putting it
inside an autonomous neural system as future research.
Moderate the title?　We will modify the title to the more specific "Can the Brain Do Backpropagation? — Exact
Implementation of Backpropagation in Predictive Coding Networks".
From Z-IL to Fa-Z-IL?　We will add the statement that Fa-Z-IL loses formal equivalence to BP, but with $t_d > 4$,
empirical equivalence always remains.
C1 in [48] should be acknowledged?　We will acknowledge this.
Clarification of [48]?　In line 42, we have stated that some previous works are equivalent to BP when feedback is
sufficiently weak (i.e., teacher weakly perturbs the network); we will add such clarification when introducing [48].
Discuss about [46]?　We will add a section outlining the differences of learning rules between [46] and Z-IL, along
which we point out that some steps may serve similar general purposes. A deeper study on the connections of the two
substantially different learning rules is left as future research.

**Reviewer #4:** Moderate the title? Will be changed to "Can the Brain Do Backpropagation? — Exact Implementation
of Backpropagation in Predictive Coding Networks".
Strong claim?　We will moderate the sentence pointed out by the reviewer.
Classification accuracy?　The classification accuracy of all situations in Fig. 3 will be added to the final paper: the
averaged accuracies of BP, IL, Z-IL, and Fa-Z-IL are 94.16%, 93.78%, 94.16%, and 94.16%, respectively.
Details of the two criteria?　The divergence of the test error is the L1 distance between the corresponding test errors,
averaged over 64 training iterations (the test error is evaluated after each training iteration). The divergence of the final
weights is the sum of the L2 distance between the corresponding weights, after the last training iteration.
Extra description of Fig. 2?　Lines 133–141 are the description of Fig. 2; this will be included in the caption of Fig. 2.

[Meta-Review · NeurIPS 2020]

Following the author response, we had a long discussion. On the positive side, this is the first algorithm with local update rules that exactly simulates BP (at least asymptotically, given complete convergence at the initialization). On the negative side, all reviewers agreed this algorithm has some reduced plausibility. Specifically, in IL (original PCN) we have to present both input and output, and wait sufficient time until convergence. In contrast, in Z-IL and Fa-Z-IL, we have to first present (only) the input, also wait sufficient time until convergence, and then present the output; In addition, the learning rule becomes more complicated (through the introduction of the Phi function) and we must detect when "the change in error node is caused by feedback input" (which seems to require some global signals). This seems more complicated and less plausible then the original IL. Another smaller issue is condition C3, which may become troublesome in a realistic continuous-time setting. Moreover, IL itself is not highly plausible. There, each weight layer is exactly duplicated through identical initial conditions and learning rules. This sounds unrealistic and possibly unstable (would the two layers diverge, given some noise or mismatch in init or dynamics?). This should be further clarified, either theoretically or empirically In the end it we were all borderline, mostly leaning towards acceptance. I strongly recommend to clarify the above issues in the camera-ready version (as well as additional issues mentioned by the reviewers), to improve the quality and impact of the manuscript.